# A Comparison of the Physiological Traits and Gene Expression of Brassinosteroids Signaling under Drought Conditions in Two Chickpea Cultivars

Khatereh Felagari [1], Bahman Bahramnejad [1,*], Adel Siosemardeh [1], Khaled Mirzaei [2], Xiujuan Lei [3,*] and Jian Zhang [4,*]

1 Department of Plant Production and Genetics, Faculty of Agriculture, University of Kurdistan, Sanandaj 6617715175, Iran; kh.felagari@uok.ac.ir (K.F.); a33@uok.ac.ir (A.S.)
2 Earth and Life Institute, University of Kurdistan, Sanandaj 6617715175, Iran; khaledmirzayi@gmail.com
3 College of Chinese Medicinal Materials, National and Local Joint Engineering Research Center for Ginseng Breeding and Development, Jilin Agricultural University, Changchun 130118, China
4 Faculty of Agronomy, Jilin Agricultural University, Changchun 130118, China
* Correspondence: b.bahramnejad@uok.ac.ir (B.B.); xiujuanl@jlau.edu.cn (X.L.); zhangjian@jlau.edu.cn (J.Z.)

**Abstract:** This study aimed to investigate the effects of drought stress at the flowering stage on the physiological and molecular responses of the genes involved in the brassinosteroid pathway of two chickpea cultivars (ILC1799: drought tolerant, and ILC3279: drought sensitive). The drought resulted in significant reductions in chlorophyll a, chlorophyll b, total chlorophyll and carotenoid content in both cultivars, and had significantly lesser effects on the tolerant cultivar, Samin, compared to that of ILC3279. However, the relative water content, the osmotic potential and the cell membrane stability were less affected by drought in both cultivars. The proline content and peroxidase activity increased significantly under drought stress in both cultivars, with a higher amount in Samin (ILC1799). Members of the *BES1* family positively mediate brassinosteroid signaling and play an important role in regulating plant stress responses. The expression of these genes was analyzed in chickpea cultivars under drought. Further, a genome-wide analysis of *BES1* genes in the chickpea genome was conducted. Six *CaBES1* genes were identified in total, and their phylogenetic tree, gene structures, and conserved motifs were determined. *CaBES1* gene expression patterns were analyzed using a transcription database and quantitative real-time PCR analysis. The results revealed that the expression of *CaBES1* genes are different in response to various plant stresses. The expression levels of *CaBES1.1*, *CaBES1.2*, *CaNAC72* and *CaRD26* genes were measured by using qRT-PCR. The relative expression of *CaBES1.2* in the drought conditions was significantly downregulated. In contrast to *CaBES1.1* and *CaBES1.2*, the expression of *CaRD26* and *CaNAC72* showed a significant increase under drought stress. The expression of *CaRD26* and *CaNAC72* genes was significantly higher in the Samin cultivar compared to that of ILC3279 cultivars.

**Keywords:** chickpea; drought stress; abiotic stress tolerance; gene expression; proline; *BES1*; *RD26*

## 1. Introduction

Chickpea (*Cicer arietinum* L.) is the second most-produced legume after soybeans worldwide [1]. Iran, with 2% of the world's total chickpea production, was the eighth-largest producer of chickpeas [1]. The chickpea is an important crop in Iran's dryland farming system, serving as a nitrogen-fixing legume in rotation with cereals. It is also a significant source of proteins, minerals, carbohydrates, fibers, and other useful compounds, especially for low-income populations [2].

Drought stress during the flowering stage is the major cause of yield loss in chickpeas [3]. Among various factors limiting chickpea growth, drought stress, especially at the end of the growing season, can cause a 33 to 45 percent reduction in yield [4]. Drought

stress is very common in semi-arid regions, such as the major part of Iran. Drought stress reduces the photosynthesis rate by changing the chlorophyll components and damaging the photosynthetic machinery [5]. Reactive oxygen species (ROS) damage chloroplasts and are the main reason for chlorophyll decrease under drought conditions [6]. Plants neutralize ROS through both enzymatic and non-enzymatic processes [7]. Enzymatic scavenging enzymes include catalase, superoxide dismutase, glutathione peroxidase, and ascorbate peroxidase, while non-enzymatic antioxidants include ascorbic acid, glutathione, and flavonoids [8]. Plants also accumulate osmolytes such as proline, glycine betaine, and carbohydrates to protect themselves against drought [9].

Brassinosteroids (BRs) are natural plant hormones essential for growth and development. They function similarly to steroid hormones in animals and play a crucial role in various processes through a complex signaling pathway [10]. Preliminary studies have shown that BRs are involved in the regulation of various physiological, biochemical, and developmental plant processes, including cell elongation in stems and roots, leaf expansion, photomorphogenesis, flowering, enzyme activation, male sterility, stomata development, and resistance to abiotic stress [11]. However, the role of BRs in drought stress is more complicated [12]. Several studies have shown that the application of exogenous BRs can increase drought tolerance in plants [13]. In contrast, a few other studies have shown that mutations in BR-related genes lead to increased drought tolerance [14,15].

BRI1-Ethylmethylsulfone-Suppressor 1 (BES1), or brassinazole resistant (BZR), is a new group of plant transcription factors activated by BRs within cells. BES1 regulates the expression of downstream BR-associated genes [16]. The BES1 genes are induced by different abiotic stresses, including drought, ABA, NaCl, and jasmonic acid [12]. It has been reported that BES1 (BZR) mediates the interaction between BR and drought signaling pathways in wheat. BES1 activates the wheat GST1 (TaGST1) and increases the drought response [17]. Moreover, BES1 targets the drought-induced transcription factor RD26 in Arabidopsis [18]. RD26 is a transcription factor that increases plant drought tolerance by promoting downstream drought-responsive gene expression [19]. RD26 is a member of the NAC family. The NAC family is one of the largest families of plant-specific transcription factors (TFs) and plays crucial regulatory roles in a wide range of developmental and stress response processes in plants.

Gene fusion expression studies have revealed that RD26 is constitutively expressed in both roots and shoots during drought and salinity stress treatments [20]. *RD26* is among the genes targeted by BES1 and is suppressed by BR signaling [12]. Drought stress-induced BR signaling blocks the transcriptional inhibition of RD26, resulting in the accumulation of RD26 proteins [12]. These accumulated RD26 proteins limit BR signaling by binding to BES1 and acting against its transcriptional activity [11].

Drought stress during the flowering stage poses a major threat to chickpea yield, causing a significant reduction in productivity. This stress is common in semi-arid regions like Iran and negatively affects photosynthesis and chlorophyll content. However, studies have shown that BRs are involved in regulating various plant processes and can enhance drought tolerance in plants. BES1, a transcription factor activated by BRs, plays a crucial role in the interaction between BR and drought signaling pathways. The regulatory mechanisms of BES1/BZR1 transcription factors in Arabidopsis and rice have been well studied [16,21]. The genome-wide identification and characterization of BES1/BZR1 genes have also been described in *Solanum tuberosum* L. [22], Chinese cabbage (*Brassica rapa* ssp. pekinensis) [23], *Glycine max* [24], *Cucumis sativus* [25], and *Beta vulgaris* [26].

Understanding the regulatory mechanisms of BES1/BZR1 transcription factors in chickpea plants will provide valuable insights into improving drought tolerance in this important legume crop. The chickpea, one of the most important legumes in the world, has a smaller genome size (730 Mb) compared to other pulse crops like lentils (*Lens culinaris* L.) and fava beans (*Vicia faba* L.) [27]. The reference genome for the chickpea species Kabuli, desi, and wild *Cicer* is accessible [28]. In the current study, we identified members of the CaBES1 gene families and CaRD26 gene homologs from the chickpea genome

based on homology, and characterized their gene structures and expression patterns under normal and drought conditions at the flowering stage. We also measured variations in key physiological indicators in response to drought stress at the flowering stage in two chickpea cultivars.

## 2. Materials and Methods

The chickpea drought-sensitive cultivar ILC3279 and the drought-tolerant cultivar ILC1799 (Samin) genotypes [29] were obtained from the office of agriculture (Jihad Keshavarzi) seed bank in Kurdistan province. To ensure their health, the seeds were treated with fungicide before being cultivated. They were planted in pots containing a mixture of farmland field, sand, and manure in a ratio of 2:2:1, with each pot having a capacity of 11 kg of soil. In each pot, seven seeds were planted at a depth of 5 cm. The experiment followed a factorial design and utilized a Randomized Complete Block Design with three replications. Drought treatment was initiated at the flowering stage, specifically when the plants reached 50% flowering. This treatment involved reducing the soils' water contents to 20% of the field capacity (FC = 20%). In contrast, the control treatment maintained humid conditions with a field capacity of 100% (FC = 100%). Until the flowering stage, all pots were irrigated once every two days. Subsequently, in the drought-treated pots, irrigation was provided at 20% of the field capacity. The field capacity was determined by monitoring the weight of the pots, which were weighed daily to calculate the field capacities.

Eight days after the drought treatments were applied at the flowering stage, leaf samples were collected for various measurements. These included assessing osmotic potential, membrane stability, relative water content (RWC), chlorophyll and carotenoid content, and peroxide enzyme activity. To preserve the samples, the leaf samples were frozen in liquid nitrogen and stored at $-80\ °C$ for subsequent gene expression analysis. For each treatment, three pots, each containing three plants, were measured.

### 2.1. Physiological Traits

To measure the relative water content (RWC), the leaves were carefully separated from each plant in the pots using a sharp knife, and their fresh weight was determined. The leaves were then placed in deionized water in the dark at room temperature for 24 h. After removing excess water, the leaves were weighed again using drying paper. Subsequently, the leaves were placed in an oven at $70\ °C$ for 48 h to determine their dry weights. The relative leaf water content was then calculated [29]. The osmotic potential of chickpea leaves in both the normal and drought conditions was determined using a KNAUER K-7400S osmometer (Berlin, Germany), following the previously described method [30].

To determine membrane stability, the electrolyte leakage method was employed [31]. Leaf samples isolated from the plants were initially washed multiple times with deionized water to remove any electrolytes present on the leaf surface. The samples were then placed in 10 mL of deionized water in a capped falcon tube and kept at room temperature (25 °C) for 24 h. After 24 h, the initial electrical conductivity (L1) was measured and recorded using an EC meter. To completely remove the leaf tissue and eliminate all electrolytes, the falcon tubes and their contents were autoclaved for 15 min. After reaching room temperature, the electrical conductivity (L2) and the electrolyte leakage rate were measured again. The electrolyte leakage percentage was determined using the formula: Electrolyte Leakage% = (L1/L2), and the membrane stability was calculated as $1 -$ electrolyte leakage $\times$ 100.

The proline content was measured as previously described [32]. Briefly, 0.2 g of leaf tissue was ground and homogenized in 4 mL of 3% (*w/v*) sulfosalicylic acid. The resulting solution was filtered using filter paper. Two milliliters of the homogenate were mixed with 2 mL of ninhydrin reagent and 2 mL of glacial acetic acid. The test tubes were then placed in a 100 °C water bath for 1 h. The samples were immediately transferred to an ice pack to stop the reaction and were then brought to room temperature. After adding 4 mL of toluene to the tubes, they were vigorously mixed for 30 s. The optical absorption of the upper solution was then measured at 520 nm.

Chlorophyll and carotenoids were extracted and measured according to Lichtenthaler (1987). The upper-third portion of three plants per treatment was ground and homogenized in 80% acetone. The absorbance was measured at 663 nm and 647 nm for chlorophyll and 470 nm for carotenoids [33].

Peroxidase activity was measured following the previously described protocol [34]. The increase in absorbance at 470 nm was monitored for 3 min, and the enzyme activity was measured in units per milligram of protein.

### 2.2. The Identification and Analysis of CaBES1 Genes

The genomes of *Cicer arietinum* were searched for BES1 genes using BLASTp with 8 protein sequences of AtBES1/AtBZR1 from *A. thaliana* as queries, with an E-value threshold of $1 \times 10^{-5}$. The obtained sequences were then analyzed using the Pfam tool to identify the conserved BES1_N domain. MEME SUIT (https://meme-suite.org/meme/ (accessed on 27 November 2023)) was used to identify conserved motifs in the BES1 proteins [35].

The molecular weights and isoelectric points of the candidate BES1 proteins were determined using Protparam (https://web.expasy.org/protparam/ (accessed on 27 November 2023)) [36]. The subcellular locations of the proteins were predicted using Plant-mPLoc.

Gene structures were determined using the GSDS 2.0 online tool (http://gsds.gao-lab.org/ (accessed on 27 November 2023)) [37], with CaBES1 and At-BES1/AtBZR1 genes and CDS sequences as inputs. The chromosomal locations of the BES1/BZR1 genes in *Medicago truncatula*, *Phaseolus vulgaris*, and *Cicer arietinum* were analyzed based on position information from the Phytozome database. The results were visualized for each species' chromosomes using the shinyCircos software. Gene duplication events were analyzed using a method described by [38]. The OrthoMCL program was used to identify gene pairs between *Medicago truncatula*, *Phaseolus vulgaris*, and *Cicer arietinum* [39].

To compare the retrieved CaBES1 sequences with AtBES1, a phylogenetic tree was built using the Molecular Evolution Genetic Analysis (MEGA) software, version X [40]. The evolutionary history was inferred using the neighbor-joining method with a bootstrap value of 1000. The genetic distance was estimated using the p-distance method.

### 2.3. Expression Analysis

The RNA-Seq data with SRA Accession numbers: SRP136396 and SRR1066056 and expression patterns of *CaBES1*-like genes in abiotic stresses were downloaded from the SRA database using SRA Toolkit's. By using TopHat alignment software, reads were aligned to the *Cicer arietinum* reference genome. The SAMtools (v0.1.19) command program was used to sort and convert the alignment data. The calculation of mapped reads and normalization was performed using BEDTools (v2.16.2) and DESeq's tools, respectively. Finally, heat maps and clusters were drawn using a ggplots package in R (v3.4.1) software from log2- (TPM+1) transformed values of *CaBES1-like* genes.

### 2.4. RNA Extraction

The total RNA was extracted from 100 mg of young leaf tissue by using an RNX-PlusTM Kit (Sinaclon) based on the supplier's protocols. For the elimination of genomic DNA, RNase-free Dnase I (Fermentas, #N0521) was used. The quantity and quality of RNA were determined using spectrophotometry via the nano-drop device (Nano-Drop Thermo Scientific–2000C, Rockford, IL, USA) at wavelengths 260 and 280 and electrophoresis on the agarose gel. The cDNA synthesis was conducted from 1.0 mg of total RNA using M-MuLV Reverse Transcriptase (RevertAid Erststrang-cDNA-Synthesekit, Thermo Scientific™, Rockford, IL, USA) using the supplier's instructions in a final volume of 20 μL.

### 2.5. Primer Design

The sequence of genes with accession numbers *CaRD26* (XM_004487622.2), *CaNAC72* (XM_004514293.2), *CaIF4A* (XM_004513380.2), *CaGAPDH* (AJ010224.1), *CaBES1.1* (XM_012719103.1) and *CaBES1.2* (XM_004500981.2) were obtained from NCBI. Primers were designed

using Primer3 (Web version 4.1.0). The sequences and characteristics of both forward and reverse primers are listed in Table 1. Primer-BLAST and Oligo Analyzer web-based software (V.3.1) were used to investigate the specificity of primers.

**Table 1.** Primer sequences used for quantitative real-time PCR.

| Gene Name | Primer Name | Primer Sequence | Tm (°C) | Amplicon Length (bp) |
|---|---|---|---|---|
| IF4 | IF4A-F | GAAGACCAACGCATTCTATCAAG | 61.6 | 128 |
| | IF4A-R | TGTTTGTGTTAGATGAGGCTGA | | |
| GAPDH | GAPDH-F | TGTCTCAGTTGTTGACCTTACAG | 60.1 | 158 |
| | GAPDH-R | CGACTTCATTGGTGATACTAGGT | | |
| NAC72 | NAC72-F | GGTTCTGTGTCGCATATACAAGA | 60.1 | 201 |
| | NAC72-R | GAATGCTTCAACAGGAGGAGAA | | |
| RD26 | RD26-F | ACACCGGCACCAAGTTAGAT | 60.1 | 168 |
| | RD26-R | GTGATTGTAGAACGTCGTCGAG | | |
| BES1.1 | BES1.1-F | AAGCCTTCTCTTCCTCGTGA | 60.1 | 198 |
| | BES1.1-R | CTCTCTCCTTCCACGTTGGT | | |
| BES1.2 | BES1.2-F | CGAGTCCAATCCATTCATACC | 60.1 | 181 |
| | BES1.2-R | GGTGATGAGATAGGCGGTGT | | |

*2.6. Real Time PCR Analysis*

Real-time PCR was conducted in an Applied Biosystems StepOne[TM] Real-Time PCR System (Rockford, IL, USA) in a volume of 15 μL containing 8.0 μL of 2× Syber Green QPCR Mix (Yektatejhiz, Iran), 1 μL of diluted cDNA, 0.5 μmol/L of both forward and reverse primers, and 5.0 μL of Rnase-free water. The PCR program was performed in the following steps: denaturation at 94 °C for 3 min, followed by 40 cycles comprising 94 °C for 20 s and 60 °C for 20 s. The PCR efficiency was calculated on five log serial dilutions for primers in both specific and reference genes. The specificity of amplification was checked using the melting curve analysis by increasing temperature from 60 to 95 °C (0.5 °C per 10 s) and gel electrophoresis. To check the genomic DNA contamination, control PCR reactions using RNA as a template were performed. In real-time PCR expression analysis, three independent biological replicates and three technical replicates of each biological sample were used. The *Ca IF4A* and *CaGAPDH* were used as reference genes to normalize and quantify gene expression. For calculating the relative expression of each gene, its Ct value was normalized to the Ct value of the reference gene [41].

*2.7. Statistical Analysis*

Both physiological and real-time expression data were analyzed using SAS version 9.1 (SAS Institute, Inc., Cary, NC, USA). A factorial two-way analysis of variance (ANOVA) with LSD post hoc test was used to reveal significant differences among treatments.

**3. Results**

*3.1. The Physiological Characters of Two Chickpea Cultivars under Drought Conditions*

The results of the physiological character analysis under drought stress showed that the RWCs of Samin and ILC3279 cultivars after about seven days of drought (20% FC) had declined by 9 and 12%, respectively, which was not significantly different compared to normal conditions (Figure 1A). The cell membrane damage examined by the electrolyte leakage of the leaf did not change significantly in both cultivars, Samin and ILC3279, under normal and drought-stress conditions (Figure 1B). Osmotic potential of the leaf also decreased in Samin, but was not significantly different from that of normal conditions. The osmotic potential in ILC3279 in drought conditions and normal conditions was similar to

that of Samin (Figure 1C). The proline content of both cultivars increased significantly with drought stress. The proline content of Samin under drought-stress conditions increased 5.1 and 10.1 times higher in Samin and ILC3279, respectively. Also, a significant difference in proline content was observed between the two cultivars in both normal and stress conditions (Figure 1D), and peroxidase activity was increased under drought conditions significantly. However, there was a significant difference between Samin and ILC3279 cultivars in both normal and drought conditions. The peroxidase activity twice in Samin and three times in ILC3279 was greater compared to the normal condition (Figure 1E).

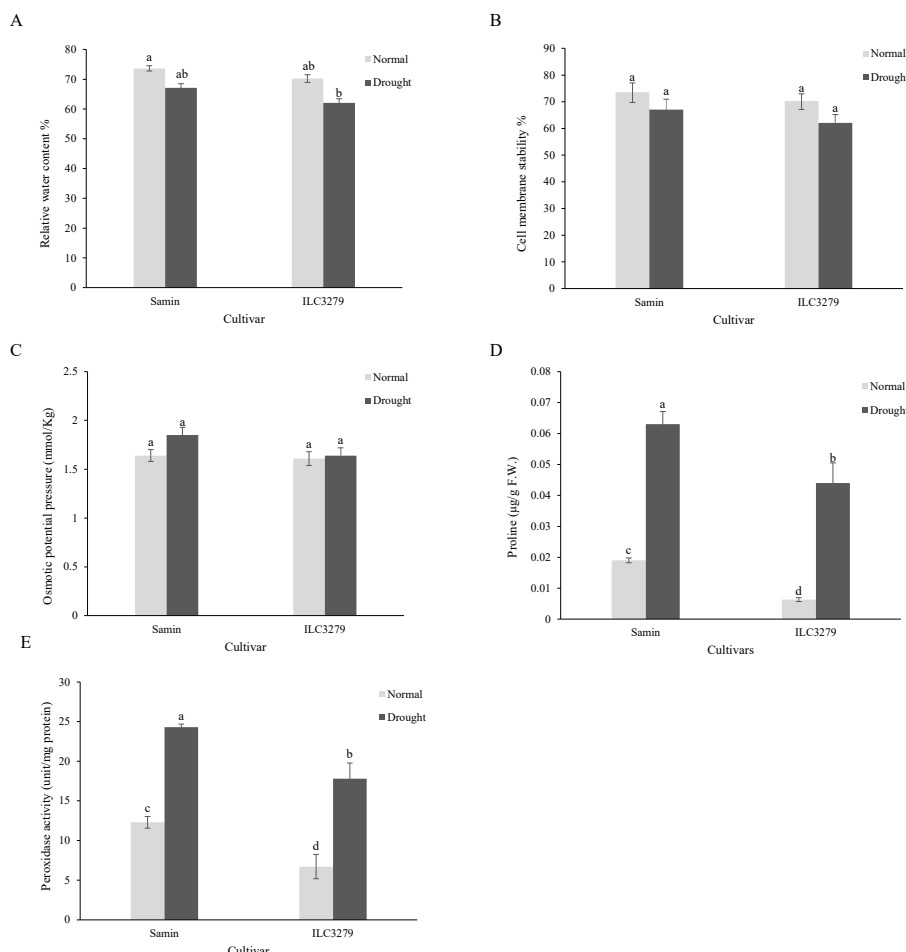

**Figure 1.** Relative water content (RWC) (**A**), cell membrane stability (**B**), osmotic potential pressure (**C**), proline content (**D**) and peroxidase activity (**E**) in the normal (FC = 100%) and drought-stressed (FC = 20%) conditions at the flowering stage in the two chickpea cultivars, Samin and ILC3279. The mean values are from three replications of each treatment with five plants for each cultivar. Different letters indicate the significance ($p < 0.05$) difference in treatments using LSD test.

Under drought-stress conditions, the chlorophyll a content significantly decreased in both Samin and ILC3279 (Figure 2A). Also, the chlorophyll a content in the Samin cultivar in both normal and drought conditions was significantly higher than that of ILC3279. The amount of chlorophyll b showed a similar pattern to that of Chlorophyll a. The total chlorophyll content decreased by 25% and 34.4% in drought conditions in Samin and ILC3279 cultivars, respectively (Figure 2B,C). The carotenoid content decreased in both cultivars under drought stress; however, in ILC3279, the decrease was significant under drought stress (Figure 2D).

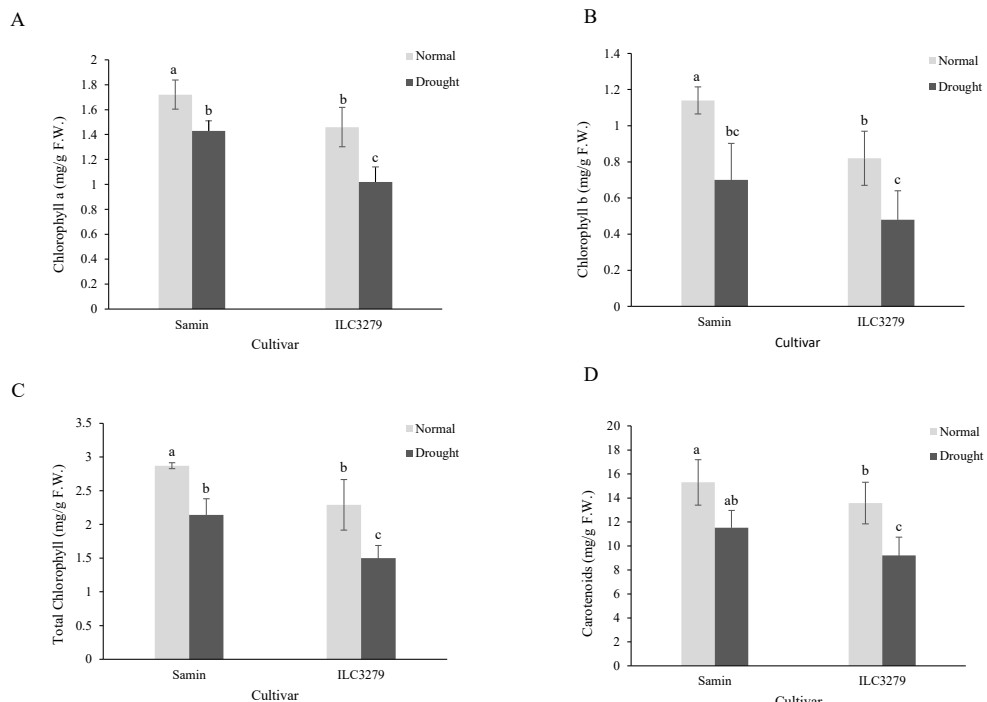

**Figure 2.** Chlorophyll a (**A**), b (**B**), total (**C**) and carotenoids (**D**) in the normal (FC = 100%) and drought-stressed (FC = 20%) conditions at the flowering stage in the two chickpea cultivars, Samin and ILC3279. The mean values are from three replications of each treatment with five plants for each cultivar. Different letters indicate the significance ($p < 0.05$) difference in treatments using LSD test.

### 3.2. The Identification of CaBES1 Genes in Chickpea Plants

To find the entire putative *CaBES1* gene, the Arabidopsis AtBES1/AtBZR1 protein sequence was used as a query to search the chickpea genome according to the obtained result. Finally, a total of six candidates of CaBES1-like genes with the usual BES1_N domain were recognized in the chickpea, and they were named *CaBES1.1* to *CaBES1.6*, in order (Table 2). The size of proteins ranged from 252 to 671 amino acids, the molecular weights varied from 26,346.96 to 7,4967.30 Da, and the predicted isoelectric points ranged from 5.45 to 9.38. According to their functions as transcription factors, all six CaBES1-like proteins were expected to be localized in the nucleus (Table 2).

**Table 2.** The basic information of *CaBES1* genes.

| Name | Gene ID | Length (aa) | Mw (Da) | pI | Chr | Location | Localization |
|------|---------|-------------|---------|-----|-----|----------|--------------|
| *CaBES1.1* | Ca_02075.1 | 271 | 29,580.08 | 9.01 | 8 | 4,822,047–4,825,123 | Nucleus |
| *CaBES1.2* | Ca_04963.1 | 316 | 34,709.73 | 9.38 | 5 | 32,553,971–32,555,767 | Nucleus |
| *CaBES1.3* | Ca_06210.1 | 671 | 74,967.30 | 5.45 | 3 | 22,988,642–22,995,557 | Nucleus |
| *CaBES1.4* | Ca_10650.1 | 650 | 73,714.08 | 5.70 | 8 | 7,413,853–7,419,598 | Nucleus |
| *CaBES1.5* | Ca_12925.1 | 321 | 34,887.85 | 8.19 | 1 | 47,282,645–47,285,551 | Nucleus |
| *CaBES1.6* | Ca_12924.1 | 252 | 26,346.96 | 5.83 | 1 | 47,290,877–47,291,866 | Nucleus |

To classify the *CaBES1-like* genes, a phylogenetic tree using the neighbor-joining algorithm for the chickpea and Arabidopsis BES1 family protein sequences was constructed (Figure 3a). There were multiple alignments of Arabidopsis and chickpea BES1 protein sequences (Figure 3b). According to the bootstrap numbers and the topology of the phylogenetic tree, six *CaBES1-like* genes were visibly clustered into two clades, A and B. Clade A contained four members (CaBES1.1, CaBES1.2, CaBES1.5, CaBES1.6) and Clade B had two members (CaBES1.3 and CaBES1.4). Each clade had a similar motif distribution and gene structure (Figures 3(aA), 4 and 5A).

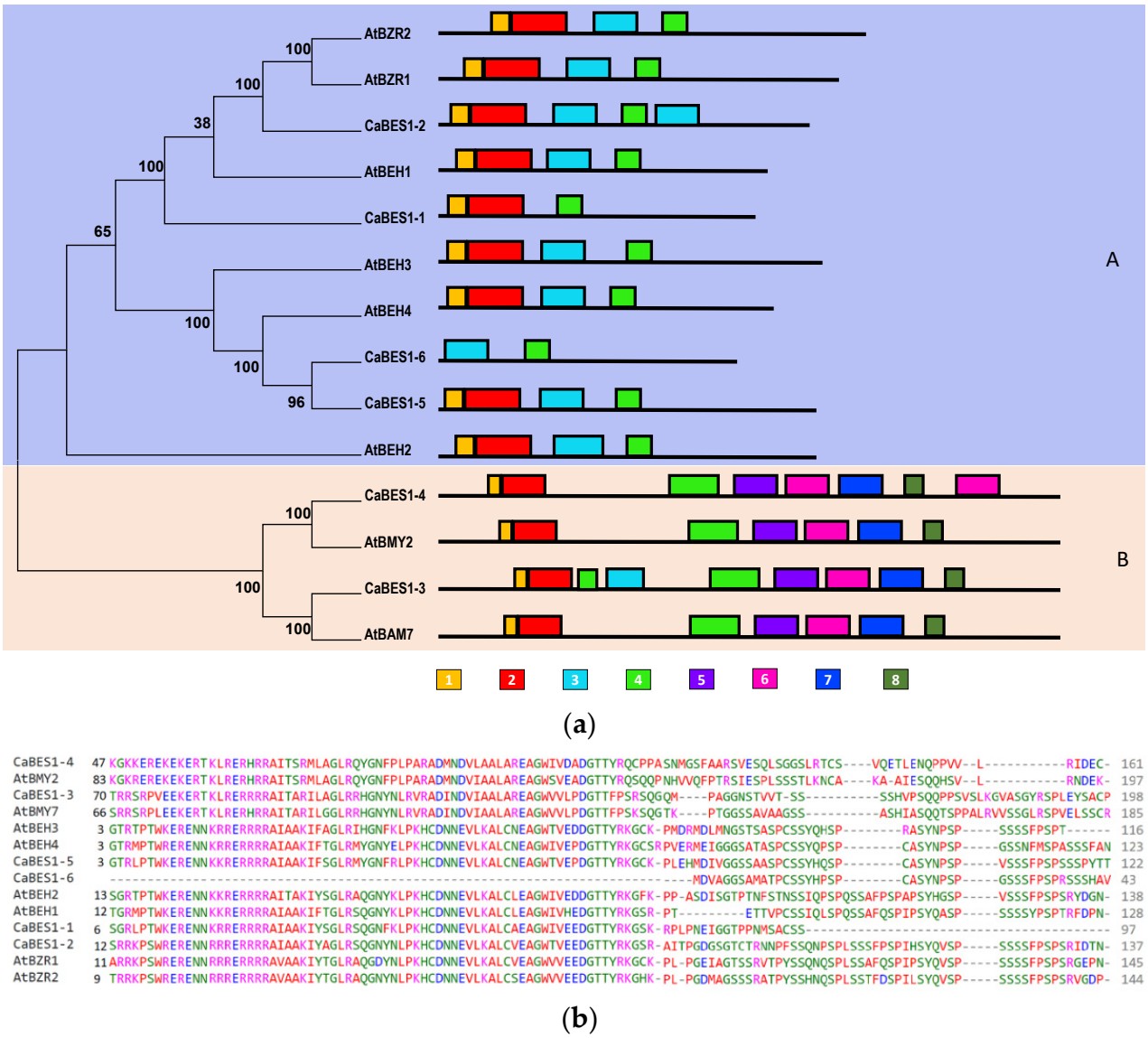

**Figure 3.** Phylogenetic and multiple alignments of BES1-like genes in Arabidopsis and chickpea. (**a**) Phylogenetic relationships and conserved motifs of BES1-like genes in Arabidopsis and chickpea using the neighbor-joining method. Clades (**A**,**B**) were shown with violet and pink, respectively. The distribution of conserved motifs of BES1-like proteins. Different motifs were indicated by different colored boxes numbered under the picture. (**b**) Multiple alignments of the N-terminal domain of chickpea and Arabidopsis BES1-like genes.

*CaBES1* genes were distributed unevenly on the eight chickpea chromosomes. The distribution of *BES1* genes on *Phaseolus vulgaris* and *Medicago truncatula* chromosomes was, indeed, uneven. *CaBES1.5* and *CaBES1.6* were located on chromosome 1, *CaBES1.3* on chromosome 3, *CaBES1.2* on chromosome 5, and the remaining two (*CaBES1.1* and *CaBES1*) on chromosome 8. All *CaBES1* gene orthologs were found on the *Phaseolus vulgaris* and *Medicago truncatula* chromosomes and have been depicted in Figure 4.

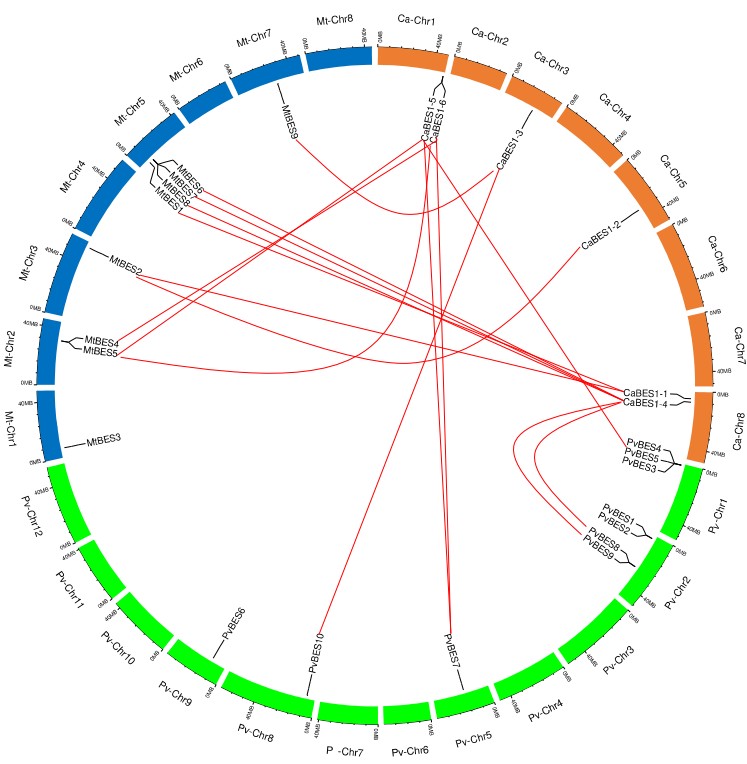

**Figure 4.** The distribution of BES1/BZR1 genes and gene pair analyses. Distribution of ZmBES1/BZR1 genes on the *Medicago truncatula* (Mt) chromosomes (blue color), *Phaseolus vulgaris* (Ph) chromosomes (green color) and *Cicer arietinum* (Ca) chromosomes (orange color). The chromosome number is indicated at the top of each chromosome. The numbers below each gene and the bottom of the chromosome indicate the gene positions and chromosome sizes. The orthologous and paralogous pairs of Ca:Mt and Ca:Pv are connected by red lines.

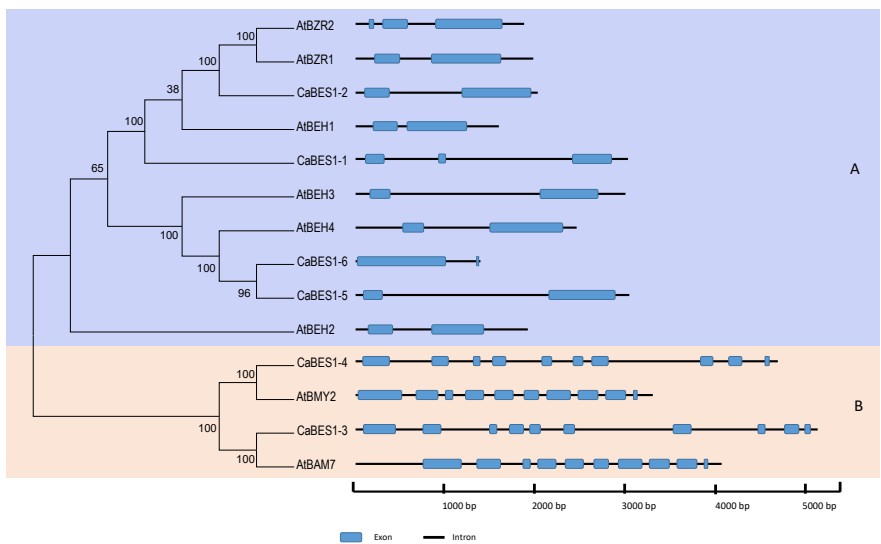

**Figure 5.** Phylogenetic analysis of BES1 genes. Phylogenetic relationships and gene structures including numbers and patterns of exons and introns in BES1 genes in Arabidopsis and chickpea. Blue balloons and black lines, respectively, show the exon and intron regions of the gene. Clades (**A**,**B**) were shown with violet and pink.

A conserved motif analysis of the obtained sequences was conducted using the MEME program and the eight motifs were predicted as is shown in Figure 3. The detail for each

motif and the result of the annotation are summarized in Table 3. The number of conserved motifs in each CaBES1 varied between two and seven. Subfamily A contained four motifs, whereas subfamily B contained seven motifs. Among the predicted motifs, two of them did not have any annotation information (Motif 1 and 8) but other motifs like 3 and 4 were predicted to contain DNA-binding transcription factor activity, and Motifs 4, 5, 6 and 7 contained beta-amylase activity. Motifs 4, 5, 6 and 7 were just observed in Clade B and Motif 3 was present only in Clade A. Motif 1 and 2 were present in all of the analyzed sequences except *CaBES1.6* (Figure 3).

**Table 3.** Conserved motifs of *CaBES1* proteins generated by MEME analysis.

| Motif Name | Discovered Motifs | E-Value | Width | Biological Process | Molecular Function | Cellular Component |
|---|---|---|---|---|---|---|
| MEME-1 | RKPTWKERENNKRRE | $2.1 \times 10^{-82}$ | 15 | None predicted | None predicted | None predicted |
| MEME-2 | RRRRAIAAKIYAGLRAQGNYNLPKHC DNNEVLKALCREAGWIVEEDGTTY | $1.5 \times 10^{-434}$ | 50 | Transcription, DNA-templated, Brassinosteroid mediated signaling pathway | DNA-binding transcription factor activity | None predicted |
| MEME-3 | GSSANASPCSSYFPSPIPSYNPSPSSSSFP SPSRSDYNNN | $6.8 \times 10^{-90}$ | 40 | Transcription, DNA-templated, Brassinosteroid mediated signaling pathway | DNA-binding transcription factor activity | None predicted |
| MEME-4 | NVDGVVVDCWWGIVEGWSPQEYNW SGYRELFNMIRDFKLKLQVVMAFHEC | $3.3 \times 10^{-88}$ | 50 | Polysaccharide catabolic process | Beta-amylase activity | None predicted |
| MEME-5 | QWVLEIGKENPDIFFTDREGRRNPEC LSWGIDKERVLRGRTGIEVYFDFM | $1.2 \times 10^{-83}$ | 50 | Polysaccharide catabolic process | Beta-amylase activity | None predicted |
| MEME-6 | EFDDFFEDGLISAVEIGLGPSGELKY PSFPEKHGWRYPGIGEFQCYDKY | $8.3 \times 10^{-70}$ | 49 | Polysaccharide catabolic process | Beta-amylase activity | None predicted |
| MEME-7 | GHTFWARGPDNAGQYNSQPHETGFF CDGGDYDGYYGRFFLNWYSQVLIDH | $2.0 \times 10^{-74}$ | 50 | Polysaccharide catabolic process | Beta-amylase activity | None predicted |
| MEME-8 | LRISNSAPVTPPLSSPTSRBP | $2.0 \times 10^{-76}$ | 21 | None predicted | None predicted | None predicted |

The analysis of gene structure revealed that sequences in Clade A had 2 to 3 exons, whereas the gene structure in Clade B differed, as all four sequences had 10 exons (Figure 5).

### 3.3. RNA-Seq Expression of BES1 Genes

The expression analysis of the *CaBES1* family was conducted using two previously reported Illumina RNA-Seq data in chickpea studies including: (A) one study with the overall design of the analysis of RNA expression in different tissue samples (root and shoot tissues of 10-day-old seedlings subjected to control (kept in water), desiccation (transferred on folds of tissue paper), salinity (transferred to a beaker containing 150 mM NaCl solution) and cold (kept in water at $4 \pm 1\ ^\circ\mathrm{C}$) stress for 5 h), and (B) another study to characterize drought adaptive strategies in two *C. reticulatum* genotypes (Savur 063 and Kalkan 064) and two chickpea cultivars, including ICC_8058 as drought-susceptible and ICC 14778 as drought-tolerant, under various drought conditions. In total, the transcription levels and patterns of the *CaBES1* gene family were considerably diverse. The expression of NAC genes *CaNAC4* (*CaRD26*), *CaNAC72*, *CaNAC55* and *CaNAC57* showed differential patterns among the control and different treatments (Figure 6). *CaBES1.1* showed constantly high levels of expression in most samples under any condition. CaBES1.6 in study A, in all the conditions, had a low expression, but in the second study, CaBES1.5 had a low expression (Figure 7). *CaRD26* showed a higher expression in roots in both the control and cold-stress conditions, as well as salinity in shoots, while *CaNAC72* showed a higher expression in shoots in both control and salinity-stress conditions. In general, all four genes showed differences in treatments. The results revealed that the expression of CaRD26 decreased in the roots under stress conditions. However, in the shoots, the expression of CaRD26 increased, particularly under desiccation stress.

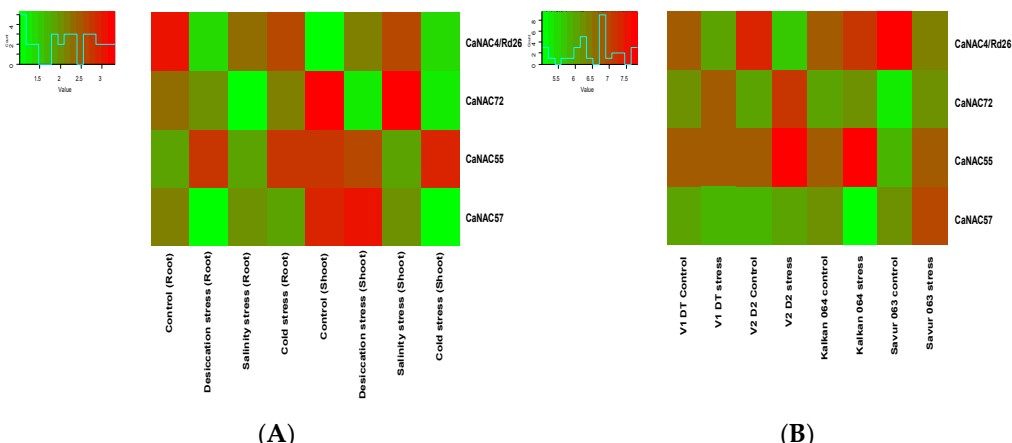

(**A**)  (**B**)

**Figure 6.** Expression patterns of four *CaNAC* genes in chickpeas in two different studies. (**A**) RNA expression in different tissue samples in 10-day-old seedlings under desiccation, salinity and cold stress for 5 h. (**B**) Analysis of gene expression in two *C. reticulatum* genotypes (Savur-063 and Kalkan-064) and two chickpea cultivars including ICC 14778 as drought-tolerant (V1-DT) and ICC_8058 as drought-susceptible (V2-DS) and under various drought conditions.

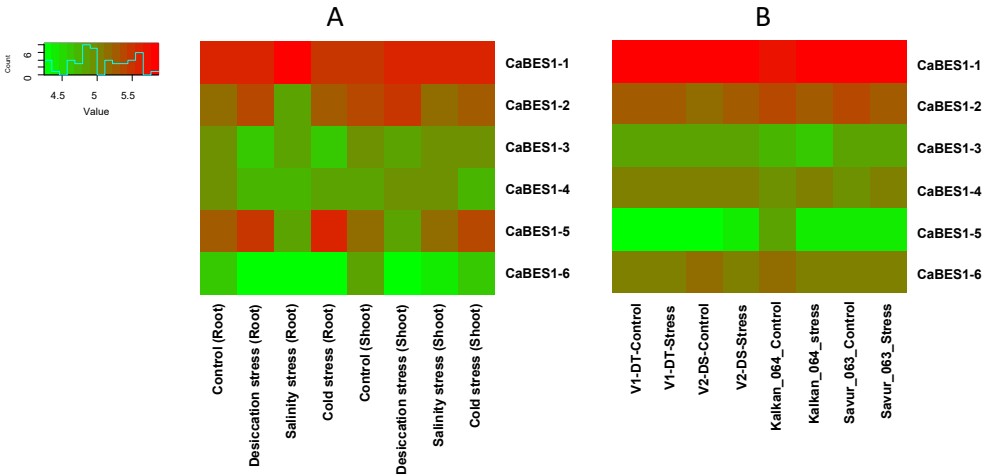

**Figure 7.** Expression patterns of *CaBES1* genes in chickpea in two different studies. (**A**) RNA expression in different tissue samples in 10-day-old seedlings under desiccation, salinity and cold stress for 5 h. (**B**) Analysis of gene expression in two *C. reticulatum* genotypes (Savur-063 and Kalkan-064) and two chickpea cultivars including ICC 14778 as drought-tolerant (V1-DT) and ICC_8058 as drought-susceptible (V2-DS) and under various drought conditions.

Furthermore, the findings demonstrated a reduction in *RD26* gene expression under drought stress in two drought-resistant (V1-DT) and drought-tolerant (V2-DS) cultivars of *C. arietiumn*. Notably, this decrease in expression was more pronounced in the sensitive cultivar (V2-DS).

### 3.4. The Expression of CaBES1.1, CaBES1.2, CaRD26 and CaNAC72 under Drought Stress

The ANOVA showed significant ($p < 0.01$) differences in the relative expression levels of the *CaBES1.1*, *CaBES1.2*, *CaRD26* and *CaNAC72* genes between cultivars and stress treatments. The *CaBES1.1* expression was differentially downregulated under drought (FC = 20%) and its expression decreased 5.7 and 7.8 times more than that of well-watered (FC = 100%) conditions in Samin and ILC3279, respectively (Figure 8A). The relative expression of *CaBES1.2* in the drought and normal conditions had a similar pattern as *CaBES1.1*, while the fold changes were 7.7 and 3.3 times reduced for *CaBES1.2* (Figure 8B). In contrast



to *CaBES1.1* and *CaBES1.2*, the expression of *CaRD26* and *CaNAC72* showed a significance increase under drought stress (Figure 8C,D). The relative expression level of *CaRD26* increased 2.2 and 3.4 times in Samin and ILC3279 under drought conditions, respectively. The expression of the *CaNAC72* gene was also different between the Samin and ILC3279 cultivars, while the fold change was lower than that of *CaRD26*.

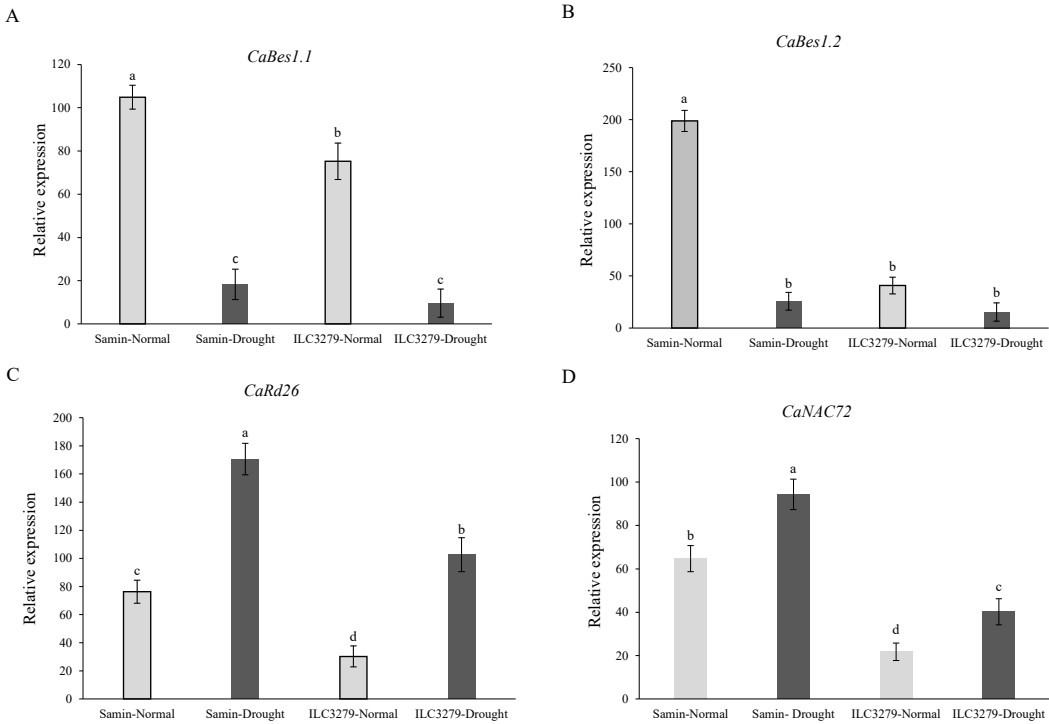

**Figure 8.** The relative expression level of the *CaBES1.1*, *CaBES1.2*, *CaRD26* and *CaNAC72* genes in the well-watered (FC = 100%) and drought-stressed (FC = 20%) conditions at the flowering stage in the two chickpea cultivars, Samin and ILC3279. Values are means with standard deviation. Different letters on bars designate a significant difference at *p* < 0.01. *CaBES1.1* expression level (**A**), *CaBES1.2* expression level (**B**), expression level of *CaRD26* (**C**), expression level of *CaNAC72* (**D**).

## 4. Discussion

In the current study, we measured some physiological traits and expressions of the genes in the brassinosteroid signaling pathways in two chickpea cultivars, Samin and ILC3279, under normal and drought conditions. ILC3279 is considered a susceptible cultivar based on its grain yield and drought-tolerance score [42]. Samin (ILC1799) showed terminal drought tolerance [42]. Among the studied physiological traits, RWC, osmotic potential and cell membrane damage were changed slightly both in cultivars under normal and drought conditions. The result of Grag et al. (2016) showed that chickpea drought-tolerant genotype ICC4958 and drought-sensitive genotype ICC1882 did not show a significance difference in RWC under drought and salinity stress conditions [43]. The authors concluded that among phenotypic traits, the root system differences and overall phenology play key roles in contrasting drought tolerance of the tolerant and sensitive genotypes.

The amount of proline was significantly higher in both genotypes under drought conditions, and in the tolerant cultivar Samin it was significantly higher in both normal and drought conditions. In chickpea cultivars, during drought stress in field conditions, both at vegetative growth and anthesis, the amount of proline was increased in the drought-tolerant cultivar ILC482 compared to that of the sensitive cultivar Pirouz [44]. In addition, proline content in both the roots and nodules of ILC3279 was increased under drought stress after the reproductive stage (before maturity) [45]. In an analysis of the proteome of two tolerant (*C. reticulatum*) and sensitive (*C. arietinum*) species under drought stress, the

amount of proline was higher in the tolerant species after drought stress [46]. Therefore, by consideration of the developmental stage and the amount of stress, the proline content maybe acted as a drought tolerant index in the chickpea. In spite of the debate on the role of proline in drought stress, it is suggested that proline has a role in free radical scavenging, the buffering of cellular redox potential and the stabilizing of cellular structure [47].

The peroxidase activity in both cultivars was significantly increased under drought conditions at the flowering stage, but in Samin it was significantly higher than that of ILC3279. Drought stress generates oxidative stress through the stomatal conductivity and reduction of internal $CO_2$, which causes the production of free radicals [48]. Accumulation ROS is one of the biochemical changes that occurs in plants under drought stress. Several reports showed that drought stress increases the amount of ROS production. Plant resistance to different environmental stresses may be related to the activity level of enzymes responsible for trapping free radicals. The POD enzyme plays a key role in detoxifying $H_2O_2$, removing malondialdehyde, which causes the peroxidation of the membrane, and maintaining the stability of the cell wall [49].

The concentration of these radicals at certain levels activates the signal transduction pathway and helps plant cells. Conversely, the high level of these radicals leads to damaging consequences on plant cells. An antioxidant enzyme such as peroxidase detoxifies these radicals. In most of the previous studies in chickpea cultivars under drought stresses, the activities of these enzymes were increased. Drought treatment in three the chickpea cultivars Bivaniej, ILC482 (both drought tolerant) and Pirouz (drought sensitive) at the flowering stage resulted in significantly higher peroxidase [49]. In the chickpea cultivar ILC-482, under control and drought treatments, the highest peroxidase activity was observed in severe stress conditions, which was almost double compared to in the non-stress treatment. Similarly, drought stress increased peroxidase activity in chickpea cultivars ICC4958, while the amount of increase in ICC4958 was significantly higher compared to that of JG315 and DCP 92-3 [50]. Drought stress increases the content of hydrogen peroxide and other reactive oxygen species and increases lipid peroxidation, and this increases the activity of antioxidant enzymes such as peroxides. In chickpea cultivars resistant to drought stress, hydrogen peroxide and lipid peroxidation levels are lower and the activity of antioxidant enzymes is seen to a greater extent.

Drought stress at the flowering stage significantly decreased the amount of chlorophyll a, b, and total chlorophyll and carotenoids contents in both cultivars. However, the amount of chlorophyll a, b, and total chlorophyll and carotenoids were higher in both the normal and drought conditions for Samin compared to ILC3279. Drought stress in chickpea cultivars usually decreases chlorophyll and carotenoids contents, but other responses, depending on the developmental stage, the duration and severity of stress and genotypes, have been observed. For example, drought stress in both vegetative and flowering stages in three cultivars, Bivaniej, ILC482 and Pirouz, significantly reduced chlorophyll content [49]. In another study, 35 chickpea genotypes were checked under normal and drought stress and the result showed that chlorophyll and carotenoid content in all genotypes were decreased by drought and drought-tolerant genotypes gathered more carotenoids than that of sensitive genotypes [51], whereas drought stress in two cultivars, Gokce and Canitez, had a different effect on chlorophyll and carotenoid contents. In drought conditions, the total chlorophyll and carotenoids did not change in Gokce, but the total chlorophyll increased in Canitez at the end of the treatment [52]. One of the major reasons for the decrease in chlorophyll content may be related to the production of free radicals and the damaging photosynthesis pigments [53].

The *BES1* gene family plays an important role in plant growth and development. Several studies have been performed in the plants; however, most of these works focus on the bioinformatics analysis of this gene. Using the sequence of *BES1* protein sequences from several plants, and also using the conservation of their domains, six highly reliable *BES1* protein sequences (E-value = 0.0001) were isolated in the chickpea genome. The *BES1* gene family has been earlier described in several plants, including Arabidopsis, cucumber,

tomato, apple, Chinese cabbage, potato, maize, and rapeseed (*Brassica napus*) [21–26]. The number of *BES1* gene families are very variable. For example, in apple, potato, maize and cucumber, the number *BES1* gene families, that have been identified are 22, 9, 11, 6, respectively. In legumes, the number of BZR gene families for pigeon pea, soybean, common bean, mung bean, chickpea and Barrel medic are 6, 16, 7, 5, 6 and 7, respectively [25].

In the current study, we conducted the expression analyses of *CaBES1* genes in *C. arietinum* using both RNA-Seq datasets and qRT-PCR. The expression levels and patterns of *CaBES1* gene family members varied considerably in different tissues, treatments and species. *CaBES1.1* exhibited a constantly high level of expression in most samples overall and *CaBES1.6* had a lower level of expression. The overexpression of *OsBES1-3* and *OsBES1-5* in rice increased the root growth of transgenic lines under drought stress, and transgenic Arabidopsis lines with overexpressed cotton *GhBES1-4* showed salt tolerance [51]. The overexpression of *PtrBES1-7* in *Populus trichocarpa* enhanced tolerance to drought stress by improving the ability to scavenge reactive oxygen species and antioxidant enzymes superoxide dismutase and peroxidase [54,55]. Previous studies showed that *BES1* genes were induced by different abiotic stresses such as drought, heat, salinity, cold and ABA [56]. The expression level of most of the ZmBES1/BZR1 genes' members in maize were induced by the ABA and the light treatments [17,57]. Also, the promoter cis-elements analysis of ZmBES1/BZR1 genes showed many classes of light-responsive elements and ABA-responsive elements (ABRE) [50]. Also, drought, salinity, cold and ABA in *Brassica rapa* induced the expression of *BrBZR* genes [58]. *BES1* and RD26 coordinate drought tolerance and plant growth in Arabidopsis [10]. Moreover, few studies showed that BES1/BZR1 in Arabidopsis might regulate the cold and drought tolerance through networking with two transcription factors RD26 (a NAC transcription activation factor) and WRKY54, or through the regulation of the transcription factors CBF, WRKY6, PYL6 and RD26 [10,57, 59,60]. Our results in two chickpea cultivars with different responses to drought stress are in agreement with that of Arabidopsis. The overexpression of the ZmBES1/BZR1-3 and ZmBES1/BZR1-9 genes in Arabidopsis decreased drought tolerance and transgenic plants displayed drought-sensitive phenotypes as well as increasing in stomatal aperture and downregulated gene expression [61]. They concluded that ZmBES1/BZR1-3 and ZmBES1/BZR1-9 negatively regulate drought tolerance via different pathways in transgenic Arabidopsis. The expression of *CaBES1* genes in both cultivars under drought stress was downregulated, while the expression of *CaRD26* was upregulated significantly. These results show that the mechanism of the *BES1* gene in interaction with RD26 in drought stress in chickpea and Arabidopsis may be similar.

## 5. Conclusions

Our results showed that the two chickpea cultivars chosen in this study respond differently to drought stress at the flowering stage. Samin, used as a drought-tolerant cultivar, produced more proline and peroxidase enzymes compared to that of ILC3279. Using a genome-wide search in chickpea, six *CaBES1* genes were identified. *CaBES1* and *CaRD26* genes were differentially regulated by various abiotic stresses, but especially by drought. Under drought conditions, the expression of *CaBES1* genes decreased, while the expression of the *CaRD26* genes increased. This comprehensive study on the effects of drought stress on physiological and molecular responses in two chickpea cultivars, Samin and ILC3279, holds significant implications for both agricultural research and practical applications. Understanding the distinct physiological traits and gene expression patterns in these cultivars under drought conditions provides valuable insights into enhancing chickpea's resilience against environmental stressors. The identified differences in chlorophyll content, proline accumulation, and peroxidase activity shed light on potential markers for drought tolerance, enabling breeders to develop more robust, stress-resistant chickpea varieties. Moreover, the in-depth analysis of the brassinosteroid pathway, particularly focusing on BES1 and RD26 genes, not only expands our knowledge of these molecular mechanisms in chickpea but also offers potential targets for genetic modification, aiming

to improve drought tolerance not only in chickpea but potentially in other crop species as well. These findings bridge the gap between fundamental plant biology and agricultural productivity, paving the way for innovative strategies to mitigate the impact of drought stress on crucial crops like chickpea.

**Author Contributions:** Data curation, K.F., K.M. and A.S.; methodology, K.F., B.B. and A.S.; software, X.L.; visualization, X.L.; writing—original draft preparation, B.B.; writing—review and editing, B.B. and J.Z. All authors have read and agreed to the published version of the manuscript.

**Funding:** This article was supported by the Jilin Agricultural University high-level researcher grant (JAUHLRG20102006). This work was financially supported by the University of Kurdistan.

**Informed Consent Statement:** Informed consent was obtained from all subjects involved in the study.

**Data Availability Statement:** Data are contained within the article.

**Conflicts of Interest:** The authors declare no conflict of interest.

## Abbreviations

| | |
|---|---|
| BR | Brassinosteroids |
| BES1 | BRI1-ethylmethylsulfone-suppressor1 |
| NAC | (NAM, ATAF and CUC) |
| RD26 | Response to desiccation 26 |
| FC | Field capacity |
| RWC | Relative water contents |
| qRT-PCR | Quantitative real-time PCR |
| EC | Electrical conductivity |
| MEGA | Molecular Evolution Genetic Analysis |

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
