# Peer review of "A Comparison of the Physiological Traits and Gene Expression of Brassinosteroids Signaling under Drought Conditions in Two Chickpea Cultivars"

_agronomy, doi:10.3390/agronomy13122963_

Round 1

Reviewer 1 Report (New Reviewer)

Comments and Suggestions for Authors

The manuscript "Comparison of physiological traits and genes expression related to drought and brassinosteroids signaling pathway in two chickpea cultivars under drought condition" of Felagaril et al. discribed the changes of some physiological parameters and gene expression between two cultivars of chickpea.

Some recomendations listed below:

1. It will be better to separate the Conclusion section and sum up the results it will intensify the information

2. It is necessary to add information on ROS generation and MDA content under drought with references. However, if the authors have this kind of data, then they can be added to the results

3. Needs the discription why only peroxidase activity was measured ? And also added short information in discussion  about activity of the main antioxidant enzymes

4. If it is posible in discussion added 1-2 more sentances about proline as a characteristic of tolerance. Perhaps the high proline accumuation in tolerate line of chickpea in compare to sensitive can be one of the marker reaction that can be used for detection of patential drough-tolerate species, lines, variety.....

Author Response

The manuscript "Comparison of physiological traits and genes expression related to drought and brassinosteroids signaling pathway in two chickpea cultivars under drought condition" of Felagaril et al. discribed the changes of some physiological parameters and gene expression between two cultivars of chickpea.

Some recommendations listed below:

  1. It will be better to separate the Conclusion section and sum up the results it will intensify the information

Conclusion was seprated.

  1. It is necessary to add information on ROS generation and MDA content under drought with references. However, if the authors have this kind of data, then they can be added to the results

We don’t have data on ROS and MDA.

  1. Needs the discription why only peroxidase activity was measured ? And also added short information in discussion  about activity of the main antioxidant enzymes.

It was added to discussion

  1. If it is posible in discussion added 1-2 more sentances about proline as a characteristic of tolerance. Perhaps the high proline accumuation in tolerate line of chickpea in compare to sensitive can be one of the marker reaction that can be used for detection of patential drough-tolerate species, lines, variety.....

We have a paragraph (391-402) in discussion section on proline accumulation in tolerant cultivars.

Reviewer 2 Report (New Reviewer)

Comments and Suggestions for Authors

Dear Author,

It is my pleasure to review the manuscript entitled “Comparison of Physiological Traits and Genes Expression Related to Drought and Brassinosteroids Signaling Pathways in Two Chickpea Cultivars under Drought Condition” a research article submitted to MDPI Journal, agronomy. Authors of this manuscript have compared two chickpea cultivars based on Physiological Traits and genes expression pattern under Drought Condition. They have performed treatment for physiological analysis and further genome-wide analysis of BES1 genes. Further confirmed by transcription database and quantitative real-time PCR analysis. Overall, the experiments, they performed, are well and the results are convincing. Thus, the presented results take up an important topic consistent with the profile of the Journal.

However, I have some suggestions, which might improve the manuscript to make important to the wider readers.

·         Few suggestions I have mentioned in the main text pdf file. Please check

·         There are many places where grammar can be improved. I suggest a careful revision by an expert.

Title: Comparison of Physiological Traits and Genes Exprerssion Related to Drought and Brassinosteroids Signaling Pathways in Two Chickpea Cultivars under Drought Condition

Suggestion: Comparison of Physiological Traits and Expression of Brassinosteroid Signaling Genes under Drought Condition in Two Chickpea Cultivars

·         Affiliation indicator of the authors should be superscript

Abstract:  

·         This study aimed to investigate the effects of drought stress at the flowering stage on the physiological and molecular responses of genes involved in the brassinosteroids pathway of two chickpea cultivars with different levels of drought tolerance (Samin: drought tolerant, ILC3279: drought sensitive).

Suggestion: This study aimed to investigate the effects of drought stress at the flowering stage on the physiological and molecular responses of genes involved in the brassinosteroids pathway of two chickpea cultivars (Samin; drought tolerant, and ILC3279; drought sensitive).

·         The drought resulted in significant reductions in chlorophyll a, chlorophyll b, total chlorophyll and carotenoid content in both cultivars, and significantly lower effects on tolerant cultivar Samin compared to that of ILC3279.------need re structured of this statement for easy understand

·         RWC---need elaboration

·         The proline content and peroxidase activity increased significantly under drought stress.---------where? In Both?

·         Why suddenly BES1 genes has been analyzed? Give a brief reason with one sentence.

1.      Introduction

·         This article lacks firm aim of the study. Aauthors should elaborate and clarify it precisely and simultaneously.

·         There are many irrelevant discussions. Introduction should be more focused on the species and key points like, Chickpea Cultivars, Physiological Traits, Genes Exprerssion, Drought and Brassinosteroids Signaling Pathways, Drought Condition.

·         L 93: Chickpea is a vital crop in the global agricultural industry, particularly in region like Iran where it plays a significant role in the dryland farming system and serves as a valuable source of nutrients for low-income populations.-------------redundant statement. Already described at first, L 37.

·         -Introduction should reflect a little results summary, which lacks here.  Rationale to be elucidated for the purpose of the study.

·         Formatting not uniform

2. Materials and Methods

·         Need references for cultivars identity. How these two varieties are considered as sensitive and tolerant? How it was developed?

·         How many pots, plants per pots, were treated? Why 8 days were treated? Need reference. What happened for phenotypic change after 8 days for both varieties?

·         Picture of phenotype change for both cultivars is important for tolerant and sensitive comparison

·         Why no morphology/phenotype has been observed? You may add this if you have data

·         How many samples were examined foe each experiment? It is necessary too.

3.  Results

·         Is there only six CaBES1 genes in chickpea? Or you have selected only this? What was the basis for this selection?

4. Discussion

Nicely presented with results sequences

However, last paragraph (L484-L504), can be used as conclusion

Comments on the Quality of English Language

Extensive editing of English language required

Author Response

Thank you for your review. Please see the attachment for my reply.

Reviewer 3 Report (New Reviewer)

Comments and Suggestions for Authors Regarding this manuscript, I provide the comments as follows:     

1) Regarding the title, the authors wrote the word "drought" twice: please revise if written it twice is necessary indeed.    1.1) Additionally, the word "Exprerssion" is not correct. "Leaf sample were" (line 124) is also incorrect, as are other phrases in the manuscript. The sentence "The expression patterns of CaBES1-like genes in abiotic stresses, RNA-Seq data with" (line 183) does not seem to make complete sense. Please revise the entire manuscript to remove minor and major English and writing errors.     

2) Throughout the manuscript, I recommend that the authors cite more articles published in 2023 to provide fresh information to the readers on this research topic.     

3) In the Abstract, I suggest that the authors provide justification for focusing on Brassinosteroids as an introductory aspect. Additionally, presenting more conclusive aspects (explaining the results obtained) in the abstract is interesting as well as introducing the plant species better.     

4) In the keywords, the authors may consider including the word "tolerance" or "abiotic stress tolerance" as well as “antioxidants”.    

5) I suggest considering including a list of abbreviations.     

6) Materials and methods: How many times did the authors conduct the experiments to increase the reliability of the obtained results?  

  6.1) I recommend that the authors mention how many technical replicates and biological replicates they used to compute and process the data for the statistical analyses.     

  6.2) Please revise the word "prepared" (line 112).    

  6.3) When it comes to biochemical analyses (e.g., regarding peroxidase), the authors cited references regarding the methods. Even so, I suggest considering providing more methodological details (for example, if there were any methodological modifications).    

    6.3.1) By the way, did the authors perform analyses regarding other enzymatic or non-enzymatic antioxidants as well?    

  6.4) The choice of the exposure time for abiotic stress treatment is not clear to me.    

  6.5) Regarding gene expression analyses:  

    6.5.1) Did the authors perform additional analyses to validate the results?     

    6.5.2) Are there more available details about the primer quality parameters?    

7) I suggest providing more perspectives for future studies (based on the results obtained) in the discussion and/or conclusion. By the way, I don't understand why there isn't a separate section for the conclusions.   

  7.1)  I recommend explaining the role of peroxidases and other antioxidants in plant cells more thoroughly in the Discussion. And once again, do the authors consider analyzing other antioxidants for the current research? Did the authors perform any omic approaches (e.g, proteomics) to increase the drough response-related insights? This might be addressed in the Discussion for further studies by the way.  

8) All gene names should be italicized. Comments on the Quality of English Language

Please see the comments to the authors.

Author Response

Regarding this manuscript, I provide the comments as follows:     1) Regarding the title, the authors wrote the word "drought" twice: please revise if written it twice is necessary indeed.    1.1)

The whole title was changed.

 Additionally, the word "Exprerssion" is not correct.

Revised.

"Leaf sample were" (line 124) is also incorrect, as are other phrases in the manuscript.

Revised.

The sentence "The expression patterns of CaBES1-like genes in abiotic stresses, RNA-Seq data with" (line 183) does not seem to make complete sense.

Revised.

 Please revise the entire manuscript to remove minor and major English and writing errors.     2) Throughout the manuscript, I recommend that the authors cite more articles published in 2023 to provide fresh information to the readers on this research topic. 

The whole manuscript was revised.    

3) In the Abstract, I suggest that the authors provide justification for focusing on Brassinosteroids as an introductory aspect. Additionally, presenting more conclusive aspects (explaining the results obtained) in the abstract is interesting as well as introducing the plant species better.     

The abstract was revised completely.

4) In the keywords, the authors may consider including the word "tolerance" or "abiotic stress tolerance" as well as “antioxidants”.  

Abiotic stress tolerance was added.

  5) I suggest considering including a list of abbreviations.   

It was added.

 6) Materials and methods: How many times did the authors conduct the experiments to increase the reliability of the obtained results?  

Revised

 6.1) I recommend that the authors mention how many technical replicates and biological replicates they used to compute and process the data for the statistical analyses. 

In real-time PCR expression analysis, three independent biological replicates and three technical replicates of each biological sample were used

   6.2) Please revise the word "prepared" (line 112).    

Revised.

6.3) When it comes to biochemical analyses (e.g., regarding peroxidase), the authors cited references regarding the methods. Even so, I suggest considering providing more methodological details (for example, if there were any methodological modifications).

It was revised.

   6.3.1) By the way, did the authors perform analyses regarding other enzymatic or non-enzymatic antioxidants as well? 

We only measured peroxidase.

  6.4) The choice of the exposure time for abiotic stress treatment is not clear to me.    

 Revised.

6.5) Regarding gene expression analyses:  

6.5.1) Did the authors perform additional analyses to validate the results?   

Yes, we used relative RT-PCR and gel quantification.

  6.5.2) Are there more available details about the primer quality parameters?

All primers checked for all parameters like primer dimer etc. In real-time PCR for each gene melting curve was checked.  Also for relative expression the primer efficiency was calculated.

 7) I suggest providing more perspectives for future studies (based on the results obtained) in the discussion and/or conclusion. By the way, I don't understand why there isn't a separate section for the conclusions.   

The conclusion section was seprated from discussion.

 7.1) I recommend explaining the role of peroxidases and other antioxidants in plant cells more thoroughly in the Discussion. And once again, do the authors consider analyzing other antioxidants for the current research? Did the authors perform any omic approaches (e.g, proteomics) to increase the drough response-related insights? This might be addressed in the Discussion for further studies by the way.  

 8) All gene names should be italicized.

Revised.

Reviewer 4 Report (Previous Reviewer 4)

Comments and Suggestions for Authors

Based on my suggestion in the first review, the authors have addressed my concerns appropriately. I still have a few questions for the manuscript.

1. The authors identified a total of six CaBES1 genes in the chickpea genome and determined their phylogenetic tree, gene structures, and conserved motifs. Could the authors also provide a chromosomal localization map for these genes to enhance the completeness of the gene information?

2. In Figure 5, please remove the role of small "b."

Comments on the Quality of English Language

Minor editing of English language required.

Author Response

The authors identified a total of six CaBES1 genes in the chickpea genome and determined their phylogenetic tree, gene structures, and conserved motifs. Could the authors also provide a chromosomal localization map for these genes to enhance the completeness of the gene information?

The chromosomal location of CaBES1 genes are presented in Fig4 for Medicago truncatula, Phaseolus vulgaris and Cicer arietinum .

  1. In Figure 5, please remove the role of small "b."

It was removed.

Minor editing of English language required.

The manuscript was revised.

Reviewer 5 Report (Previous Reviewer 1)

Comments and Suggestions for Authors

Although the author has made some changes, the experimental data are still not accurate, and there are no innovative findings. In addition, there is not enough experimental evidence to prove the drought resistance of these two cultivar materials drought-resistant material Samin and drought-sensitive material ILC3279. In addition, the change trends and fold change of the two materials were almost the same under drought stress(figures 1 and 2), and there was no significant difference. Moreover, there were no phenotypic photos of plants under drought stress in the manuscript, which also led to doubts on the experimental materials. Therefore, I still insist on my comments and reject for publishing of this manuscript.

Comments on the Quality of English Language

Author Response

Thank you for reviewing the manuscript, please see the attachment.

Round 2

Reviewer 2 Report (New Reviewer)

Comments and Suggestions for Authors

It is good, and improved well. Table one format should be changed. If there is no bar from journal for fig. number, you may add plant pic in the text.

Reviewer 3 Report (New Reviewer)

Comments and Suggestions for Authors

The authors have addressed my previous recommendations. I have no more comments.

This manuscript is a resubmission of an earlier submission. The following is a list of the peer review reports and author responses from that submission.

Round 1

Reviewer 1 Report

Comments and Suggestions for Authors

The study compares the differential physiological characteristics under drought stress between the drought-tolerant variety (Samin) and the drought-sensitive variety (ILC3279). Additionally, it investigates the expression variations of BES genes among different varieties under drought conditions using qPCR. However, the paper lacks novelty and the research content is fundamental. The manuscript writing requires enhancement, as there are numerous errors in the details. For instance, in Figure 1A, there is a misspelling in the y-axis label, "wter" should be corrected to "water"; in Figure 1D, the unit for Proline should be μg/g; in line 159, there is an error in the temperature unit, it should be 95instead of 95 C. Furthermore, the text size in the figures is too small, making it difficult to read (e.g., Figure 1, Figure 2, Figure 6, Figure 7, etc.). Most significantly, the analysis of gene expression among different drought resistance varieties did not yield any meaningful results.

Comments on the Quality of English Language

Moderate editing of English language required

Reviewer 2 Report

Comments and Suggestions for Authors

1、There is a spelling mistake in Figure 1.

2、The author's name in reference 13 is incorrect.

3、The Latin species name in the reference should be italicized.

Comments on the Quality of English Language

English language fine.  Minor editing of English language required.

Reviewer 3 Report

Comments and Suggestions for Authors

The study conducted by Felagari et al is definitely an interesting study that is needed to improve crop production. However, the study suffers frommvarious reporting issues which are outlined below.

L38-39 and 43-44: Include a references for the statements.

L46: what does ROS stand for?

L53-55: The sentence is not clear. Please rewrite.

L74-76: include a reference.

L80-84: Be consistent in your style. Either write all crops in their english name or scientific name. If you are writing in scientific name, make sure they are italicized. Same goes to the rest of the manuscript.

Conclude the introduction section by stating the rationale of the study not just what will be done in the study.

L103-106: The last part of the sentence is not clear.

L130: change "one" to "1"

One of my major concerns in this study is the incomplete presentation of methods used. More details are needed. Authors have not mentioned the settings used in the various software programs as well as their references. These are very important for reproducibility of research.

The result section need serious editing and rewrite. Authors need to take their time and make sure the results are properly presented and interpreted. Some of the interpretations are not right. Authors should look into this. The figure labelings is alos different from the in-text citations. 

Table 1 can be moved to the methods section. 

The presentations of the methods and result made me lose interest in the discussion and conclusion. If the result is not well interpreted, then the discussion will have issues. 

Finally, authors should conclude by stating the implication of their study.

Comments on the Quality of English Language

The manuscript should be thoroughly checked by an english language editor. It will benefit greatly from this.

Reviewer 4 Report

Comments and Suggestions for Authors

In this manuscript, authors investigated the effects of drought stress at the flowering stage on the physiological and molecular responses of genes involved in the brassinosteroids pathway of two chickpea cultivars with different levels of drought tolerance. I found this topic interesting but I have many concerns related to the manuscript. There are still good rooms to improve the quality of the manuscript. I recommend the authors to have all my comments addressed and revise the manuscript.

Major Revisions:

1. I did not find any description of the results for Figure 5 in the results section of this paper. Kindly ask the authors to review this.

2. In section 3.3, the process by which the NAC genes CaNAC4 (CaRD26), CaNAC72, CaNAC55, and CaNAC57 were obtained has not been adequately explained. While these genes were briefly introduced in the introduction, there is no subsequent analysis provided. Their sudden appearance in the results section is quite abrupt and lacks continuity.

3. In section 3.4, are "CaBES1.1" and "CaBES1.2" referring to "CaBES1-1" and "CaBES1-2," respectively? Furthermore, this article contains numerous errors, such as failing to italicize Latin names and missing paragraphs. As it stands, this academic paper is not suitable for publication in its current form.

4. Line 370-395, the discussion merely consists of descriptions of the results, which lack depth. I suggest that the authors should compare and discuss the findings with drought indicators from other species for a more comprehensive analysis.

5. The title of the article mentions “genes related to the brassinosteroid signaling pathway”, but I believe that this paper is more suitable for an analysis of the BES1 gene family and its role under drought stress. The current title seems too broad in scope.

6. In the earlier sections, the authors measured physiological indicators of two chickpea varieties under drought stress. However, the subsequent gene analysis seems disconnected from these indicators, resulting in a lack of coherence in the main body of the article. It would be beneficial if the authors could address this issue and enhance the relevance between the physiological measurements and the subsequent gene analysis.

Minor Revisions:

1. Are BES1 and BES1/BZR the same type of transcription factor? The author sometimes refers to BES1 and sometimes to BES1/BZR. This inconsistency is confusing and should be standardized. For instance, in lines 65 and 68.

2. Line 85, if RD26 is indeed a gene, then what you have identified are its homologous genes rather than a gene family.

3. In section 2.2, the utilization of certain software tools lacks proper citation of references, such as, shinyCircos software, MEGA.

4. There is a sequencing error in the order of Figures 6 and 7 as provided by the author.

Comments on the Quality of English Language

The English language of this article should undergo professional proofreading before publication.